# Synthesis of Brominated Lactones Related to Mycalin A: Selective Antiproliferative Activity on Metastatic Melanoma Cells and Inhibition of the Cell Migration

**DOI:** 10.3390/md21060349

**Published:** 2023-06-07

**Authors:** Domenica Capasso, Paola Marino, Sonia Di Gaetano, Nicola Borbone, Monica Terracciano, Roberta Trani, Caterina Longo, Vincenzo Piccialli

**Affiliations:** 1CESTEV, University of Naples Federico II, 80145 Naples, Italy; domenica.capasso@unina.it; 2Department of Chemical Sciences, University of Naples Federico II, 80126 Naples, Italy; paola.marino2@studenti.unina.it; 3Institute of Biostructures and Bioimaging, CNR, 80134 Naples, Italy; 4Department of Pharmacy, University of Naples Federico II, 80131 Naples, Italy; nicola.borbone@unina.it (N.B.); monica.terracciano@unina.it (M.T.); 5Department of Bioscience, Biotechnology and Environment, University of Bari “Aldo Moro”, Via Orabona 4, 70125 Bari, Italy; roberta.trani@uniba.it (R.T.); caterina.longo@uniba.it (C.L.)

**Keywords:** mycalin A lactone, bromo-lactone analogues, antiproliferative activity, A375, HeLa and WM266 cells, wound healing assays

## Abstract

Starting from D-xylonolactone and D-ribonolactone, several five-membered bromolactones, related to the C1–C5 portion of mycalin A lactone, have been synthesized. The bromination of D-ribonolactone with HBr/AcOH, without a subsequent transesterification step, has been studied for the first time, giving us most of the acetylated lactones investigated in the present study. For each compound, where possible, both the C-3 alcohol and the corresponding acetate were prepared. Evaluation of their anti-tumor activity showed that all the acetates possess a good cytotoxicity towards human melanoma (A375), human cervical adenocarcinoma (HeLa) and human metastatic melanoma (WM266) cancer cells, comparable or even higher than that displayed by the original mycalin A lactone. Lactone acetates derived from D-ribonolactone showed the higher selectivity of action, exhibiting a strong cytotoxicity on all the tested tumor cells but only a limited toxicity on healthy human dermal fibroblast (HDF) cells, used as a control. Wound healing assays showed that two of these substances inhibit the migration of the WM266 cells.

## 1. Introduction

Mycalin A (**1**, Figure 1) is a polybrominated C_15_ acetogenin [1] isolated some years ago in our group from the encrusting sponge *Mycale rotalis* [2,3]. As a continuation of our interest in the search for new anti-tumor substances, either of synthetic or natural origin [4,5,6], we recently reported that mycalin A displays strong antiproliferative activity on human melanoma (A375) and human cervical adenocarcinoma (HeLa) cells (Figure 1 and Table 1) [7]. Nevertheless, mycalin A also showed a strong cytotoxicity towards human dermal fibroblasts (HDF) used as healthy control cells (Figure 1 and Table 1).

To select structural analogues of the natural substance that could retain the original anti-tumor activity but which showed a reduced cytotoxicity towards healthy cells, various analogues of mycalin A were synthesized [7]. Among the synthesized compounds, mycalin A acetate **2** (Figure 1) showed an increased cytotoxicity towards all the tested cancer cells compared to mycalin A (Table 1). However, this substance was still very active against the healthy HDF cells (Table 1). Lactone **3** (Figure 1), henceforward called mycalin A lactone, lacking the C1–C3 side chain of mycalin A acetate **2**, proved to be one of the most active substances among the prepared analogues. In addition, this substance displayed a toxicity towards HDF cells ten times lower than that shown towards the A375 cells (Table 1). 

In this paper, we report the synthesis of several simplified bromolactone compounds mimicking the C1–C5 portion of mycalin A lactone **3** (Figure 2), which possess diverse substitution and/or configuration at the C-2, C-3 and C-4 carbons. The rationale underlying this study assumes that the activity and selectivity of compound **3** could be mostly attributed to its lactone portion. D-xylono-1,4-lactone and D-ribono-1,4-lactone were chosen as the starting materials. Indeed, previous studies [7] have shown that the THF-containing or THF-mimicking portion of mycalin A, such as lactone **3**, is essential to its biological activity because its absence or its perturbation cause a strong decrease in cytotoxic activity.

The evaluation of the cytotoxic activity of the synthetic substances was performed on the A375 and HeLa cancer cell lines, previously used for the assays on mycalin A and its lactone derivative **3**. Due to our interest in studying metastatic progression [8,9], the highly metastatic human melanoma cells (WM266) were also tested in this study. Human dermal fibroblasts (HDF) were used as healthy cells to evaluate the selectivity of action of the investigated compounds towards cancer cells.

## 2. Results and Discussion

Starting from D-xylonolactone [10], six five-membered bromolactones were synthesized (Figure 3); both the C-3 alcohol and the corresponding acetates were prepared. They all retain the C-3 configuration of D-xylonolactone. First, we synthesized the bromolactone **4** (Figure 3), which closely matches the lactone portion of **3**. Our purpose was to clarify whether the lactone moiety in **3** might be responsible for its activity.

Thus, D-xylose was converted into D-xylonolactone by treatment with Br_2_ under basic conditions followed by quenching with formic acid. Next, according to a reported procedure [10], the latter was treated with 33% HBr/AcOH (HBA) and then with MeOH to give dibromolactone **5** (Figure 1) as the main product (31% from D-xylose), which possess an inverted configuration at C-2 with respect to D-xylose. Selective hydrogenolytic debromination of **5** at C-2 was then accomplished with Pd/C (10%) in EtOAc to give alcohol **6** in a 62% yield. Finally, acetylation of **6** with Ac_2_O in the presence of catalytic amounts of HClO_4_ afforded lactone **4** in an 89% yield (17% from D-xylonolactone, four steps). Purification by HPLC gave a sample of **4**, which is suitable for the biological assays.

It is to be noted that 3% of the C-2 epimeric dibromolactone **7** (Figure 1) was also obtained from the bromination of D-xylonolactone. Compounds **5** and **7** were separated by HPLC. NMR data for **5** agree with those reported [10]. As expected for a five-membered ring, the inversion of configuration at C-2 in **7** affected the multiplicity of the H-2 and H-3 signals that appear as a singlet and a doublet, respectively, due to the absence of coupling between H-2 and H-3 (these protons appear as a double doublet and a double double doublet in the proton spectrum of **5**) caused by the near 90° dihedral angle between the planes in which the H(2)-C(2)-C(3) and H(3)-C(3)-C(2) atoms lie. Although the NMR data for **7** were substantially like those reported in the literature [11], some signals displayed a non-negligible chemical shift difference. Therefore, we decided to confirm the structure of this compound by X-ray diffraction analysis. The formation of **7** strongly suggested that a tautomerization process involving the carbonyl lactone of **5** had occurred under the acidic reaction conditions. As far as we know, the formation of this substance had not previously been observed during the bromination of D-xylonolactone [10].

As for the proton spectrum of **6**, the disappearance of the one-proton signal for H-2, resonating at 4.80 ppm in the proton spectrum of **5**, and the appearance of an AB-system further coupled in the range 2.6–3.0 ppm are clear evidence of the absence of the C-2 bromine in **6**. Similar NMR features characterize the acetate **4**. Furthermore, the downfield shift of the H-3 signal at δ 5.58 (H-3 resonates at δ 4.73 in the proton spectrum of **6**) and the presence of a three-proton singlet at δ 2.13 in the proton spectrum of **4** clearly indicate the acetylation of H-3.

It is known that lactones such as **5** are prone to epimerize at C-2 under basic conditions [12]. Therefore, to obtain higher amounts of **7**, compound **5** was treated with Et_3_N in EtOAc for 1 h (Figure 2). Indeed, under these conditions, a 1:1 mixture of **7** and unreacted **5** was obtained. Compound **7** had previously been synthesized from D-lyxonolactone [12] and from potassium D-lyxonate [11,13]. Alcohols **5** and **7** were acetylated under the same conditions used to acetylate **6**, giving **8** and **9**, respectively, in 68% (borsm) and 66% yields (Figure 2).

The ^1^H-NMR features of the acetates **8** and **9** are substantially like those displayed by the corresponding alcohols **5** and **7**, respectively. The presence of acetate signals (**8**: δ 2.20; **9**: δ 2.16) and a low-field shift of the H-3 protons in their proton spectra are observed in these cases as well (see their proton spectra in Appendix A for further details).

The antiproliferative activity of the compounds **4**–**9** was then evaluated on the A375, HeLa and WM266 cells (Figure 4). As for the acetates **4**, **8** and **9**, all exhibited an activity higher than that shown by the corresponding alcohols **5**–**7** towards all the tested tumor cells, both at 1 μM and 10 μM concentrations. This result is in line with that observed for mycalin A and some of its acetate/alcohol derivative pairs, previously synthesized [7], and can be ascribed to the easier cell penetration of the acetates, caused by their reduced polarity. In particular, compound **4** induced a good inhibition of the proliferation of the Hela and WM266 cells of about 65% at 10 μM concentrations, while, at the same concentration, a strong 80% inhibition of the A375 cells was observed.

Interestingly, both the C-2 brominated epimeric substances **8** and **9** exhibited a cytotoxic activity higher than that displayed by **4** on all the tested cells, particularly on the A375 and WM266 cells (Figure 4), for which a cytotoxic effect of about 90% and 80%, respectively, was observed. To evaluate the selectivity of the acetates **4**, **8** and **9** towards tumor cells, the inhibition of the healthy HDF cells was then assessed for these compounds (Figure 4). It was observed that the lactones **8** and **9** were the most selective compounds, showing a limited killing effect towards the HDF cells (**8**: 50%; **9**: 40%). Compound **4**, on the contrary, showed a rather limited selectivity causing a 70% death of the HDF cells, a value which is not unlike that shown against the HeLa and WM266 tumor cells, but is higher than that shown by the original lactone **3** on the HDF cells that displayed a 30% killing effect [7].

As a next step, the IC50 values for the most selective compounds **8** and **9** were evaluated on the WM266 highly metastatic melanoma cells. The cytotoxicity of the original lactone **3** towards WM266 cells, not previously assayed for this substance, was also evaluated for comparison purposes. To this end, compound **3** was synthesized from mycalin A as described [7]. As previously observed for the Hela and A375 cells (Table 1), lactone **3** displayed a good activity towards the WM266 cells and a more limited killing effect of the HDF cells (Figure 5 and Table 2). Gratifyingly, compounds **8** and **9** displayed IC_50_ values lower than that of lactone **3**, while a reduced (for **9**) or similar (for **8**) killing effect was observed on the HDF cells, with respect to **3** (Table 2).

The promising data obtained for the xylonolactone-derived lactones prompted us to enlarge the collection of lactone analogues of type **4**–**9** by synthesizing further representatives of this class of compound. It was interesting to study the effect of the configuration at C-3 on the cytotoxicity. Hence, several lactones possessing an inverted configuration at that stereogenic centre were synthesized, starting from D-ribose.

In this context, it is to be noted that the bromination of D-xylonolactone (Figure 1) only leads to the 4R isomer **5**, and consequently, only its 4R derivatives **4** and **6**–**9** (Figure 1 and Figure 2) could be obtained from it. On the other hand, the bromination of D-ribonolactone is reported [11,14,15] to give the three isomeric dibrominated compounds **10** (2S,4S, major), **11** (2R,4S, minor), and **12** (2S,4R, minor) (Figure 6). Therefore, this process was expected to provide us with three out of the four possible isomers of **5** involving the C-2/C-4 centres, allowing us to study the effect of the stereochemistry of the lactone ring on the antiproliferative activity.

Overall, thirteen five-membered lactones, most of which obtained as C-3 alcohol/acetate pairs, were obtained (Figure 6) either by the direct bromination of D-ribonolactone or by further synthetic manipulation of the substances obtained from that process. In particular, the isomeric C2/C4 dibromo compounds **10**–**12** and **13**–**15**, their C-2 debrominated analogues **19**–**22**, the two C2-C3 unsaturated compounds **17** and **18** and the C-5 debrominated lactone **16** were synthesized according to the procedures reported below. Several of these acetates had not previously been synthesized.

D-ribonolactone [16] was synthesized from D-ribose by treatment with Br_2_ in alkaline solution, according to a described procedure [17] and obtained in a pure form by crystallization from EtOH. Then, its treatment with 33% HBA gave compound **10** as the main product (34%) along with minor amounts of its C-2 and C-4 epimers **11** (4%) and **12** (12%), respectively (Figure 3), which were separated by HPLC.

Some NMR features of the isomeric alcohols **10**–**12** deserve a brief comment. The ^1^H-NMR data for **10** well agree with those reported in the literature [11]. While an accumulation of the signals from H-2 to H-4 protons in the range 4.50–4.65 is seen in its proton spectrum, a very good dispersion of these signals, in the range 4.34–4.72 and 4.25–5.0 ppm, is observed for its C-2 and C-4 epimers **11** and **12**, respectively. The proton spectrum of the latter is identical to that of its enantiomer **7** (See Appendix A).

According to the same protocol used to synthesize alcohol **4** (Figure 1), compound **10** was subjected to the hydrogenolytic debromination at C-2 to give alcohol **19** in 76% yield (Figure 4), which was acetylated, leading to acetate **20** in 75% yield (60% for two steps) [11].

The proton spectrum of **19** includes signals at δ 2.58 and δ 2.99 for the H-2 protons, (AB system further coupled), which indicated the absence of the bromine at C-2. Similarly, the proton spectrum of **20** includes signals for the H-2 protons at δ 2.62 and δ 3.13 (AB system further coupled) and a three-proton singlet at δ 2.11 for the acetate.

Both **19** and **20** were subjected to cytotoxicity assays (Figure 7 and Appendix A). As observed for the analogous pair of compounds **4** and **6** of the D-xylose series (Figure 1), acetate **20** proved to be by far more active than the corresponding alcohol **19** on all three tested cell lines. On the other hand, a comparison of the biological data of **20** and **4** indicated that the inversion of configuration at C-3 in the former had caused an increased cytotoxic activity on the A375 and WM266 cells (Figure 7), bringing the cell death to approximately 90% for both. On the contrary, a very slight increase in the cytotoxicity towards the HeLa cells was observed for **20** with respect to **4**. Importantly, an insignificant killing effect on the HDF cells was observed for this compound. Indeed, compound **20** is one of the most active substances among those prepared (see also Figure 8 below) and, considering the data for the HDF cells (Figure 7), the one showing the highest selectivity of action towards tumor cells.

Compound **19** had previously been obtained from D-ribonolactone in a 47% overall yield in four steps, according to a slightly different protocol [16]. The acetylation of **19** to **20** was reported by Bock et al. [11] to proceed in a rather low yield (33%). In our hands, this transformation gave **20** in a higher (75%) yield, and consequently, combining the reported yield for **19** [16] and our yield for its acetylation, compound **20** could be obtained, in principle, in five steps and a 35% overall yield, starting from D-ribonolactone.

Having observed that for the **4**/**6** and **19**/**20** pairs, the acetates displayed an activity higher than that of the corresponding alcohols, we decided to carry out the bromination of D-ribonolactone without the successive transesterification with MeOH. It was expected that the acetyl-derivatives **13**–**15** corresponding to the alcohols **10**–**12**, could be directly obtained, thus rendering their preparation easier. We also aimed at studying the possible minor products (if any) of this process to enlarge the collection of the lactone substances to be assayed. As far as we know, the bromination of D-ribonolactone had previously been carried out with the sole intent of obtaining alcohols **10**–**12** and the transesterification step was needed to achieve this end.

Therefore, D-ribonolactone was treated with 33% HBA for four days (Figure 5) and the reaction mixture was carefully separated by preparative HPLC on a silica column. In addition to the expected acetates **13**–**15**, the three further lactones **16**–**18** were obtained (overall yield of **13**–**18** 42%). The structural correspondence between **10**–**12** and **13**–**15** was proven by the acetylation of the former to the latter. In addition, the NMR data for **13**–**15** were in good agreement with those reported in the literature [11]. The NMR spectra of acetate **15** are identical to those of its enantiomer **9**.

Compound **13** had previously been obtained in an unspecified yield by the acetylation of the corresponding alcohol **10**, in turn prepared in a 36% yield by reaction of D-ribonolactone with HBA for 60 h according to the two-step procedure shown in Figure 3 [11]. Compounds **14** and **15** had not previously been synthesized. The corresponding alcohols **11** and **12** were obtained as minor products (1% and 7% yields, respectively) from the reaction of D-ribonolactone with HBA [11].

Compound **16** had previously been obtained only in one instance as a 6:1 mixture with **13** [18,19] by the reaction of D-ribonolactone with HBA, followed by the reaction with Ac_2_O, and used without purification for further synthetic elaborations. The low-field (60 MHz) ^1^H-NMR data reported for **16**, obtained from the **16**/**13** mixture, were limited to the CH_2_OAc group and did not give conclusive information on the structure of **16**. Therefore, a complete spectral characterization of **16** was carried out and its structure was confirmed by the acetylation of the corresponding diol, in turn synthesized from D-ribonolactone as described [11]. The proton spectrum of diacetate **16** is strongly reminiscent of that of **13** (see Appendix A), the only noticeable difference being the downfield shift of the H_2_-5 protons (**13**: δ 3.74; **16**: δ 4.41) due to the presence of an acetyl group at C-5 in place of the bromine.

Unsaturated lactone **17** had not previously been synthesized. The corresponding alcohol was obtained in a 30% yield as a side-product in one of the steps of the synthesis of some allylic bromides from 1,2:5,6-diisopropylidene- D mannitol [20]. Compound **18** had previously been synthesized [21] in a racemic form and an 8.5% yield from the €-pentadienoic acid. The NMR data of **18** perfectly agree with those reported [21]. The proton spectra of unsaturated compounds **17** and **18** are very similar to each other (see Appendix A). Both include a narrow doublet (**17**: 7.48 ppm; **18**: 7.59 ppm; *J* = 1.8 Hz for both) for the H-3 vinyl proton. The presence of an acetate at C-5 in **17** induces a downfield shift of the signals for H_2_-5 that appears as a narrow AB system further coupled, centred at 4.34 ppm. These signals appear in the proton spectrum of **18** as two well-separated signals in the range 3.4–3.7 ppm.

We suppose that the formation of **17** and **18** could be due to the loss of acetic acid from the corresponding acetates **16** and **13**, respectively, during the work-up or purification steps. Indeed, we have observed that compound **13** tends to lose AcOH in various acidic conditions such as on standing in CDCl_3_, or on silica gel. Additionally, the acetylation of **10** to give **13** (with Ac_2_O/HClO_4_ (cat.)) is accompanied by the formation of a certain amount of **18**. Finally, even when stored dry at 4 °C, **13** slowly transforms into a mixture of its C-2 epimer **14** and the compound **18**. A 4:2:1 mixture of **13**, **14** and **18** was formed from **13** after about three months at 4 °C. On the contrary, the C-2 epimeric compound **14** is stable for months at 4 °C and only a 5% amount of both **13** and **18** was formed after prolonged treatment with HBA at room temperature, while no trace of either **13** or **14** is observed when **18** was reacted with HBA under the same conditions for three days. Interestingly, compound **9**, the xylonolactone-derived C-3 epimer of **13**, has no tendency to lose AcOH and can be stored for months at 4 °C. We presume that the loss of acetic acid in **13**, as well as its C-2 epimerization to give **14**, might occur through the enol intermediate **23** (Figure 6).

Finally, the C-2 debrominated compounds **21** and **22** were obtained from **12** in 25% and 35% yields, respectively (Figure 7), according to the procedure used to synthesize **4**. Compound **21** had previously been obtained from the corresponding β-hydroxy-acid by treatment with *N*-bromosuccinimide [22]. The acetate **22** had never previously been synthesized. The NMR data reported for **21** were limited to those pertaining to the proton spectrum collected at 60 MHz. Therefore, a complete high-field characterization of **21** was carried out. NMR spectra of **21** and **22** were identical to those of the corresponding enantiomers **6** and **4**, respectively.

The results of cytotoxicity assays on the acetates **10**–**18** and **22** are shown in Figure 8. Some observations about their activity and selectivity with respect to their xylose-derived isomers **4**, **8** and **9** are worth making. Cytotoxicity assays on the corresponding alcohols **10**–**12** and **21** are reported in the Appendix A (see Appendix A).

First, a comparison of the activity of the three stereoisomers **4**, **20** and **22**, all strictly similar to the C1–C5 portion of mycalin A lactone, seems important. As far as the enantiomeric compounds **4** and **22** are concerned, the latter displays an increased activity towards the A375 and especially towards the WM266 cells compared to **4**, while essentially no difference can be noticed for these compounds on the HeLa cells. Importantly, contrary to **4**, which shows a killing effect of about 70% towards the HDF cells, compounds **20** and **22** are essentially inactive towards these cells. Thus, compounds **20** and **22**, which possess with respect to **4** the inversion of configuration at C-3, or at both C-3 and C-4, respectively, are the most active and selective of the three stereoisomers **4**, **20** and **22**. This suggests that the S configuration at C-3 could play an important role for the selectivity.

As for compounds **13**–**15**, and their xylonolactone-derived stereoisomers **8** and **9**, they all possess an additional bromine at C-2 with respect to the compounds **4**, **20** and **22**. They all display a high activity, not too far from that exhibited by their debrominated analogues, mostly towards the A375 and WM266 cells (Figure 8). Compound **13** is in general slightly less active than the other lactones belonging to this group but displays a selectivity like that shown by the other two ribonolactone-derived isomers **14** and **15**. It is to be also noted that once again, the ribonolactone-derived substances are the least active against the healthy cells (**13**–**15** vs. **8** and **9**), thus paralleling the behavior of the C-2 debrominated substances.

It is also interesting to compare the cytotoxicity of compounds **13** and **16**, which only differ in the substituent at the C-5: a bromine in **13** and an OAc group in **16**. Compound **16** is more active than **13** on all the cell lines at the 10 μM concentration but is much less selective. This suggests that the presence of a bromine at C-5 might play an important role in the increase in the selectivity. Indeed, **16** is the least selective compound in the ribose series at a 10 μM concentration. Interestingly, it is still very active and selective towards WM266 cells at a 1 μM concentration (Figure 8, WM266: 60% killing effect; HDF: 20% killing effect). This evidence and its potential easy preparation in high amounts make it a preferential target for future studies.

Finally, the unsaturated lactones **17** and **18** are both very active and selective towards all the cell lines, although they are functionalized in a different way with respect to all the other lactones, possessing a C2/C3 double bond and no acetate group at C-3. In this case, the different substitution at C-5 (**17**: OAc; **18**: Br) seems not to affect their biological behavior.

All the above observations deserve further in-depth studies. In this study, we have selected compounds **18** and **20**, two of the most promising substances, for further studies on the WM266 cells. The dose–response curves for these compounds and their IC_50_ values are shown in Figure 9 and Table 3, respectively. The IC_50_ values stand at around 1–2 μM for both compounds on the WM266 cells, while they are about thirty times higher for the HDF cells.

Since metastasis formation involves cell migration, an in vitro scratch assay was performed on WM266 cells to investigate whether **18** and **20** could interfere with this mechanism. After 24 h of seeding, the cell monolayers were scratched linearly and incubated with **18** and **20** at a concentration corresponding to half of their IC_50_ values. This choice was dictated by the need to avoid a high mortality of the cells assayed at 48 h, the end point of this assay. The concentrations chosen were 0.85 µM and 0.6 µM, for **18** and **20**, respectively. The results obtained show that in the presence of both **18** and **20** the wound healing was delayed compared to the control (Figure 10). Remarkably, this assay demonstrated that **20** is the most effective substance on the inhibition of WM266 migration.

Finally, our attention was focused on the improvement of the synthesis of **18**, which had previously been prepared only in a racemic form and in an 8.5% yield [22]. It was reported that various 2-substituted γ-lactones possessing a leaving group at C-3 undergo a facile elimination under weakly basic conditions to give the corresponding 2-substituted α,β-unsaturated lactones [23]. Therefore, the xylose-derived lactone **8** (Figure 2), which can be easily synthesized from D-xylonolactone in high amounts, was chosen as a suitable starting material from which compound **18** could be obtained through the C2/C3 elimination of AcOH (Figure 8). However, in our hands the use of various basic conditions (Et_3_N in EtOAc or EtOAc/H_2_O, 95:5; NaHCO_3_ in MeOH or MeOH/H_2_O, 95:5) did not lead to the desired compound **18**.

The attention was then addressed to the use of NaHSO_3_. This decision was based on the previous observation that a brominated 1,4-lactone possessing *trans* diaxially oriented C-2 hydrogen and C-3 acetoxy groups, strictly similar to **8**, transformed quantitatively into the corresponding butanolide via a NaHSO_3_-induced (NaHSO_3_ in MeOH) *trans* elimination of AcOH [15]. This observation also suggested that the butanolide product could be stable to the action of NaHSO_3_. However, it should be noted that the experimental conditions under which this transformation was performed are not reported in the literature [15]. Thus, treatment of **8** with an equimolar amount of NaHSO_3_ in MeOH for 2 h gave **18** in a 30% yield (Table 4, entry 1). Prolonged reaction times (22 h) only resulted in a decreased yield of **18** (25%, Table 4, entry 2). The reaction in the presence of excess NaHSO_3_ (10 eq.) in MeOH raised the yield of **18** to 45% (Table 4, entry 3). The addition of water to the mixture (MeOH/H_2_O, 95:5) allowed a reduction in the required amount of NaHSO_3_ to 1.2 eq., giving **18** with the same 45% yield, in a reduced reaction time (1.5 h) (Table 4, entry 4). 

## 3. Materials and Methods 

### 3.1. General Experimental Procedures

All reagents were purchased (Aldrich, St. Louis, MO, USA) at the highest commercial quality and used without further purification. The reactions were monitored using thin layer chromatography (TLC) carried out on precoated silica gel plates (Merck 60, F254, 0.25 mm thick, Rahway, NJ, USA) using a KMnO_4_ solution as the stain. Merck silica gel (Kieselgel 40, particle size 0.063–0.200 mm) was used for the column chromatography. All the HPLC separations and purifications were performed on Phenomenex Luna columns (25 cm × 10 mm or 25 cm × 4.6 mm, 5 µm), using *n*-hexane/EtOAc mixtures as eluent, at flow rates of 2.5 or 1 mL/min., respectively. Na_2_SO_4_ was used as a drying agent for aqueous work-up. Nuclear magnetic resonance (NMR) experiments were acquired on Bruker Avance Neo 600 and 700 MHz spectrometers (Bruker-Biospin, Billerica, MA, USA) or Varian Unity Inova 500 MHz spectrometer (Palo Alto, CA, USA) in CDCl_3_. The NMR spectra were processed using the MestReNova (version 14.3.0, Mestrelab Research, Santiago de Compostela, Spain) suite. Proton chemical shifts were referenced to the residual CHCl_3_ signal (7.26 ppm). ^13^C-NMR chemical shifts were referenced to the solvent (CDCl_3_, 77.0 ppm). Coupling constants (*J*) are given in Hertz. Abbreviations for signals multiplicities are as follows: s = singlet, d = doublet, t = triplet, q = quartet, m = multiplet, b = broad. IR spectra were recorded on a FT-Ir Nicolet 5700 instrument. Abbreviations: br = broad; s = strong. The optical rotations were measured using a JASCO P-2000 polarimeter at the sodium D line. The high-resolution mass spectra were recorded by infusion on a Thermo Linear Trap Quadrupole (LTQ) Orbitrap XL mass spectrometer equipped with an electrospray source in the positive mode using MeOH as the solvent.

### 3.2. Bromolactones **4**–**9**

#### 3.2.1. Synthesis of Lactone **5**

The following is a slightly modified procedure with respect to that reported in the literature [10].

D-xylose (1.0 g, 6.66 mmol) was dissolved in water (2.7 mL) and the solution was cooled to 5 °C (ice-bath). Then, K_2_CO_3_ (1.13 g, 8.18 mmol) was added in portions under stirring, maintaining the temperature below 20 °C. To the clear solution, bromine was dropwise added (1.23 g, 400 μL); during the addition, the temperature was maintained in the range 5–10 °C. The cloudy, yellow-orange mixture was stirred for 30 min at this temperature and then overnight at room temperature. The process was quenched by addition of formic acid (65 μL) and the mixture was stirred for 20 min. Acetic acid (20 mL) was added and the aqueous suspension was taken to dryness under an air stream at 45–50 °C, furnishing a syrupy product. ^1^H-NMR analysis confirmed the formation of the D-xylono-1,4-lactone [10].

Acetic acid (3 mL) was added to the crude D-xylono-1,4-lactone obtained as above, followed by 33% HBr/AcOH (5.2 mL), and the mixture was stirred at 45 °C for 1 h and then at room temperature for 2 h. The TLC analysis (hexane-EtOAc, 1:1) showed the formation of mainly two products at *R*_F_ = 0.5 (major product) and *R*_F_ = 0.6 (minor product) attributable to the alcohol **5** and the corresponding acetate **8**, respectively.

MeOH (8.5 mL) was added in portions to the above mixture at 0 °C over a period of 30 min. After stirring for two days at room temperature, TLC analysis showed the complete conversion of the acetate **8** into the corresponding alcohol **5**. The mixture was filtered through a Büchner funnel to remove KBr, giving a clear solution. The MeOH was evaporated under reduced pressure and the mixture was co-evaporated twice with water (2 × 3 mL). The resulting solid was partitioned between water and EtOAc and the aqueous phase was extracted twice with EtOAc. The combined organic phases were dried, filtered, and taken to dryness. Residual acetic acid was azeotropically removed with *n*-heptane (2 × 5 mL) furnishing a brown oil (1.16 g). Column chromatography on silica gel, eluting with *n*-hexane-EtOAc (2:8), gave alcohol **5** (564 mg) contaminated by a small amount (by ^1^H-NMR) of **7** (**5**: 31%; **7**: 3%). Pure compounds **5** and **7**, suitable for biological assays, were obtained by HPLC (analytical silica column, mobile phase: *n*-hexane-EtOAc, 65:35).

**5** (3S,4S,5S)-3-bromo-5-(bromomethyl)-4-hydroxydihydrofuran-2(3H)-one: [α]_D_^20^ = –19.7 (c = 1.59, CHCl_3_). IR (neat) ν_max_ 3420 (br), 1770 cm^−1^. ^1^H-NMR [10,24] (500 MHz, CDCl_3_): δ 4.80 (1H, d, *J* = 4.4, H-2), 4.69 (1H, m, H-4), 4.63 (1H, bdd, *J* = 3.5, 3.5, H-3), 3.73 (1H, A part of an AB system further coupled, *J =* 10.1, 8.7, Ha-5), 3.67 (1H, B part of an AB system further coupled, *J* = 10.1, 5.8, Hb-5), 2.60 (1H, br, OH). ^13^C-NMR: (125 MHz, CDCl_3_): [25] δ 169.4, 80.3, 68.8, 48.2, 25.9. HRESIMS (High resolution electrospray ionization mass spectrometry) *m*/*z*: calcd for C_5_H_6_^79^Br_2_NaO_3_ 294.8581 [M + Na]^+^, found: 294.8566 [M + Na]^+^, 296.8547 [M + Na + 2]^+^, 298.8523 [M + Na + 4]^+^.

**7** (3R,4S,5S)-3-bromo-5-(bromomethyl)-4-hydroxydihydrofuran-2(3H)-one: [α]_D_^20^ = −7.80 (c = 1.73, CHCl_3_). IR (neat) ν_max_ 3420 (br), 1770 (s) cm^−1^. ^1^H-NMR (500 MHz, CDCl_3_): δ: 4.99 (1H, dt, *J* = 7.5, 7.5, 3.4, H-4), 4.66 (1H, d, *J* = 3.4, H-3), 4.24 (1H, s, H-2), 3.67 (2H, d, *J* = 3.4, H_2_-5), 3.4–2.7 (1H, br, OH). ^13^C-NMR: (125 MHz, CDCl_3_): δ 171.2, 80.6, 73.9, 41.1, 25.5. HRESIMS *m*/*z*: calcd for C_5_H_6_^79^Br_2_NaO_3_ 294.8581 [M + Na]^+^, found: 294.8570 [M + Na]^+^, 296.8548 [M + Na + 2]^+^, 298.8530 [M + Na + 4]^+^.

#### 3.2.2. Alcohol **6**

Pd/C (10% *w*/*w*, 8.0 mg) and Et_3_N (30 μL) were added to compound **5** (53.0 mg, 0.193 mmol) in EtOAc (1.5 mL), and the mixture was stirred under a hydrogen atmosphere. A vacuum-fill technique was employed using a hydrogen balloon and a three-way vacuum adapter. After 1 h, the reaction mixture was filtered over celite, and the filtrate was taken to dryness under reduced pressure to give a mixture of the starting product **5** and the corresponding C-2 debrominated lactone **6** (39.4 mg). Column chromatography on silica gel (eluent EtOAc) gave unreacted **5** (16.2 mg) and the desired compound **6** (23.4 mg, 62%), as a collarless oil. Further HPLC purification on an analytical silica column (eluent: *n*-hexane-EtOAc, 3:7) afforded a sample of **6** suitable for biological assays.

**6** (4R,5S)-5-(bromomethyl)-4-hydroxydihydrofuran-2(3H)-one: [α]_D_^20^ = −23.9 (c = 0.27, CHCl_3_). IR (neat) ν_max_ 3430 (br), 1770 (s) cm^−1^. ^1^H-NMR (500 MHz, CDCl_3_): δ 4.73 (1H, bdd, *J* = 5.6, 3.8, H-3), 4.61 (1H, m, H-4), 3.69–3.62 (2H, m, H_2_-5), 2.84 (1H, dd, *J* = 17.9, 5.6, Ha-2) 2.65 (1H, d, *J* = 17.9, Hb-2), 2.4–2.0 (1H, broad, OH). ^13^C-NMR: (125 MHz, CDCl_3_): δ 174.7, 82.1, 67.7, 38.6, 26.4. HRESIMS *m*/*z*: calcd for C_5_H_7_^79^BrNaO_3_ 216.9476 [M + Na]^+^, found: 216.9461 [M + Na]^+^, 218.9449 [M + Na + 2]^+^.

#### 3.2.3. Acetate **4**

A 65% HClO_4_ solution (two drops) was added to alcohol **6** (11.8 mg, 0.0605 mmol) in acetic anhydride (1 mL). After 30 min, TLC analysis (*n*-hexane-EtOAc, 2:8) showed the complete conversion of **6** (*R*_F_ = 0.5) into a less polar product (*R*_F_ = 0.7). Ice was added and the mixture was extracted with CH_2_Cl_2_ (3 × 3 mL). The organic phase was washed with water, dried and taken to dryness. Filtration through a short pad of silica gel gave crude **4** (14.5 mg) as an oil. Further purification by HPLC on an analytical silica column (*n*-hexane-EtOAc, 65:35) gave pure acetate **4** (12.7 mg, 89%) as a white solid which crystallized on standing.

**4** (2S,3R)-2-(bromomethyl)-5-oxotetrahydrofuran-3-yl acetate: [α]_D_^20^ = −47.8 (c = 0.21, CHCl_3_). IR (neat) ν_max_ 1796 (s), 1736 (s), 1237 (s) cm^−1^. ^1^H-NMR (500 MHz, CDCl_3_): δ 5.58 (1H, ddd, *J* = 5.8, 4.4, 1.1, H-3), 4.76 (1H, ddd, *J* = 8.3, 5.9, 4.3, H-4), 3.59 (2H, AB system further coupled, A part: *J* = 10.3, 5.9; B part: *J* = 10.3, 8.3), 2.93 (1H, dd, *J* = 18.3, 6.0, Ha-2), 2.67 (1H, dd, *J* = 18.3, 1.1, Hb-2), 2.13 (3H, s, acetate). ^13^C-NMR: (175 MHz, CDCl_3_): δ 173.0, 169.6, 80.3, 69.3, 36.8, 25.7, 20.7. HRESIMS *m*/*z*: calcd for C_7_H_9_^79^BrNaO_4_ 258.9582 [M + Na]^+^, found: 258.9560 [M + Na]^+^, 260.9539 [M + Na + 2]^+^.

#### 3.2.4. Lactone **7**

Et_3_N (four drops) was added to a stirred solution of **5** (5.5 mg, 0.0201 mmol) in EtOAc (1 mL). After 1h, the reaction mixture was taken to dryness and the residue was filtered on a short pad of silica gel eluting with EtOAc to give 5.3 mg of an oil. TLC analysis showed that it was essentially composed by two products at *R*_F_ = 0.6 and *R*_F_ = 0.65. HPLC separation on an analytical silica column (*n*-hexane/EtOAc, 8:2) furnished the starting products **5** (2.5 mg) and its C-2 epimeric alcohol **7** (2.5 mg).

#### 3.2.5. Acetate **8**

A 65% HClO_4_ solution (two drops) was added to alcohol **5** (22.8 mg, 0.0832 mmol) in acetic anhydride (0.5 mL). After 1 h, TLC analysis (*n*-hexane-EtOAc, 1:1) showed the almost complete conversion of **5** (*R*_F_ = 0.4) into a less polar product (*R*_F_ = 0.6). Usual work-up gave a white solid (22.3 mg). Further separation by analytical HPLC (*n*-hexane-EtOAc, 8:2) gave the starting product **5** (3.0 mg) and acetate **8** (15.1 mg, 68% borsm), as a crystalline solid.

**8** (2S,3S,4S)-4-bromo-2-(bromomethyl)-5-oxotetrahydrofuran-3-yl acetate: [α]_D_^20^ = −61.9 (c = 0.78, CHCl_3_). IR (neat) ν_max_ 1786 (s), 1758 (s), 1197 cm^−1^. ^1^H-NMR (500 MHz, CDCl_3_): δ 5.90 (1H, dd, *J* = 5.4, 4.4, H-3), 4.83 (1H, m, H-4), 4.79 (1H, d, *J* = 5.5, H-2), 3.63 (1H, A part of an AB system further coupled, *J* = 10.2, 6.2, Ha-5), 3.67 (1H, B part of an AB system further coupled, *J* = 10.2, 8.8, Hb-5), 2.20 (3H, s, acetate). ^13^C-NMR: (125 MHz, CDCl_3_): δ 168.9, 168.7, 78.8, 69.0, 42.8, 25.4, 20.3. HRESIMS *m*/*z*: calcd for C_7_H_8_^79^Br_2_NaO_4_ 336.8687 [M + Na]^+^, found: 336.8685 [M + Na]^+^, 338.8662 [M + Na + 2]^+^, 340.8643 [M + Na + 4]^+^.

#### 3.2.6. Acetate **9**

Alcohol **7** (20.0 mg, 0.0730 mmol) was acetylated under the same conditions used for **6**. After 1 h, TLC analysis (*n*-hexane-EtOAc, 6:4) showed the complete conversion of **7** (*R*_F_ = 0.4) into a less polar product (*R*_F_ = 0.5). Usual work-up followed by filtration through a short pad of silica gel gave crude **9** (24.1 mg). Further purification by analytical HPLC (*n*-hexane-EtOAc, 8:2) afforded acetate **9** [11] (15.1 mg, 66%) as an oil.

**9** (2S,3S,4R)-4-bromo-2-(bromomethyl)-5-oxotetrahydrofuran-3-yl acetate: [α]_D_^20^ = +9.1 (c = 0.71, CHCl_3_). IR (neat) ν_max_ 1780 (s), 1750 (s), 1226 (s), 1187 (s) cm^−1^. ^1^H-NMR (500 MHz, CDCl_3_): δ 5.47 (1H, d, *J* = 3.4, H-3), 5.11 (1H, m, H-4), 4.24 (1H, s, H-2), 3.64 (1H, A part of an AB system further coupled, *J* = 10.1, 5.7, Ha-5), 3.55 (1H, B part of an AB system further coupled, *J* = 10.1, 9.0, Hb-5), 2.16 (3H, s, acetate). ^13^C-NMR: (125 MHz, CDCl_3_): δ 169.5, 169.0, 78.6, 74.7, 38.3, 24.8, 20.4. HRESIMS *m*/*z*: calcd for C_7_H_8_^79^Br_2_NaO_4_ 336.8687 [M + Na]^+^, found: 336.8676 [M + Na]^+^, 338.8654 [M + Na + 2]^+^, 340.8634 [M + Na + 4]^+^.

### 3.3. Bromolactones **10**–**22**

D-ribonolactone was synthesized starting from D-ribose according to a reported procedure [15] and obtained in pure form by crystallization from EtOH. Its purity was confirmed by ^1^H and ^13^C-NMR spectra.

To D-ribonolactone (526 mg, 3.55 mmol), obtained as above, 33% HBr/AcOH (3.0 mL) was added, and the mixture stirred at r.t. for four days. Then, one third of the reaction mixture was partitioned between water and CHCl_3_ and the aqueous phase was extracted with CHCl_3_ (3 × 5 mL). The combined organic phases were dried, filtered and taken to dryness. Residual acetic acid was azeotropically removed with *n*-heptane (2 × 2 mL), to give an oil that was filtered through a short pad of silica gel eluting with EtOAc and then EtOAc/MeOH in (95:5). The combined eluates were taken to dryness furnishing an oily product (167.2 mg). Separation by HPLC on a silica column (250 × 10 mm), using *n*-hexane-EtOAc (7:3) as the mobile phase gave compounds **13** (22.3 mg, 6%), **14** (11.2 mg, 3%), **15** (11.1 mg, 3%), **16** (41.7 mg, 12%), **17** (24.9 mg, 9%), and **18** (25.0 mg, 9%).

MeOH (5 mL) was added in portions to the remaining original reaction mixture in a water bath. After 16 h, the reaction mixture was taken to dryness under an air stream at 50 °C. The resulting oily semi-solid product was partitioned between water and CHCl_3_ and the aqueous phase was extracted with CHCl_3_ (3 × 5 mL). The combined organic phases were dried, filtered and taken to dryness. The residual acetic acid was azeotropically removed with *n*-heptane (2 × 3 mL). Separation of the crude by HPLC on a silica column (250 × 10 mm), using n-hexane-EtOAc (6:4) as the mobile phase, gave the slightly impure alcohol **10** (91.0 mg) and pure **11** (9.4 mg, 4%) and **12** (27.0 mg, 12%). The further purification of **10** by HPLC on an analytical silica column, using n-hexane-EtOAc (75:25) as the mobile phase, furnished pure **10** (82 mg, 34%).

**10** (3S,4R,5S)-3-bromo-5-(bromomethyl)-4-hydroxydihydrofuran-2(3H)-one: [α]_D_^20^ = +14.5 (c = 1.21, CHCl_3_). IR (neat) ν_max_ 3445 (br), 1785 (s) cm^−1^. ^1^H-NMR (500 MHz, CDCl_3_): δ 4.63 (1H, dd, *J* = 6.7, 5.8, H-3), 4.54 (1H, ddd, *J* = 5.8, 5.8, 4.8, H-4), 4.52 (1H, d, *J* = 6.7, H-2), 3.72 (1H, A part of an AB system further coupled, *J* = 11.4, 5.7, Ha-5), 3.67 (1H, B part of an AB system further coupled, *J* = 11.4, 4.8, Hb-5). ^13^C-NMR: (150 MHz, CDCl_3_): δ 169.4, 82.3, 78.0, 44.3, 29.9. HRESIMS *m*/*z*: calcd for C_5_H_6_^79^Br_2_NaO_3_ 294.8581 [M + Na]^+^, found: 294.8590 [M + Na]^+^, 296.8571 [M + Na + 2]^+^, 298.8549 [M + Na + 4]^+^.

**11** (3R,4R,5S)-3-bromo-5-(bromomethyl)-4-hydroxydihydrofuran-2(3H)-one: [α]_D_^20^ = +16.2 (c = 0.23, CHCl_3_). IR (neat) ν_max_ 3450 (br), 1784 (s) cm^−1^. ^1^H-NMR (700 MHz, CDCl_3_): δ 4.71 (1H, d, *J* = 6.0, H-2), 4.59 (1H, ddd, *J* = 5.4, 4.1, 4.1, H-4), 4.36 (1H, ddd, *J* = 6.8, 5.7, 5.7, H-3), 3.76 (1H, A part of an AB system further coupled, *J* = 11.9, 4.2, Ha-5), 3.67 (1H, B part of an AB system further coupled, *J* = 11.9, 3.9, Hb-5), 2.51 (1H, d, *J* = 6.8, OH). ^13^C-NMR: (175 MHz, CDCl_3_): δ 169.0, 81.7, 70.2, 45.9, 29.9. HRESIMS *m*/*z*: calcd for C_5_H_6_^79^Br_2_NaO_3_ 294.8581 [M + Na]^+^, found: 294.8579 [M + Na]^+^, 296.8561 [M + Na + 2]^+^, 298.8542 [M + Na + 4]^+^.

**12** (3S,4R,5R)-3-bromo-5-(bromomethyl)-4-hydroxydihydrofuran-2(3H)-one: [α]_D_^20^ = +3.2 (c = 0.27, CHCl_3_). IR (neat) ν_max_ 3432 (br), 1778 (s) cm^−1^. ^1^H-NMR (500 MHz, CDCl_3_): δ 4.99 (1H, dt, *J* = 7.2, 7.2, 3.4, H-4), 4.66 (1H, dd, *J* = 3.4, 1.1, H-3), 4.25 (1H, bd, *J* = 1.1, H-2), 3.68 (2H, d, *J* = 7.2, H_2_-5). ^13^C-NMR: (125 MHz, CDCl_3_): δ 171.4, 80.7, 73.9, 41.1, 25.5. HRESIMS *m*/*z*: calcd for C_5_H_6_^79^Br_2_NaO_3_ 294.8581 [M + Na]^+^, found: 294.8563 [M + Na]^+^, 296.8543 [M + Na + 2]^+^, 298.8526 [M + Na + 4]^+^.

**13** (2S,3R,4S)-4-bromo-2-(bromomethyl)-5-oxotetrahydrofuran-3-yl acetate: IR (neat) ν_max_ 1796 (s), 1748 (s), 1217 cm^−1^. ^1^H-NMR (700 MHz, CDCl_3_): δ 5.49 (1H, dd, *J =* 3.2, 3.2, H-3), 4.68 (1H, ddd, *J =* 6.2, 6.2, 3.1, H-4), 4.43 (1H, d, *J =* 3.4, H-2), 3.74 (2H, AB system further coupled; A part: *J* = 11.5, 5.5; B part: *J* = 11.5, 4.7, H_2_-5), 2.17 (3H, s, acetate). ^13^C-NMR: (175 MHz, CDCl_3_): δ 169.5, 169.2, 82.4, 77.8, 38.5, 30.0, 20.6. HRESIMS *m*/*z*: calcd for C_7_H_8_^79^Br_2_NaO_4_ 336.8687 [M + Na]^+^, found: 336.8690 [M + Na]^+^, 338.8670 [M + Na + 2]^+^, 340.8649 [M + Na + 4]^+^.

**14** (2S,3R,4R)-4-bromo-2-(bromomethyl)-5-oxotetrahydrofuran-3-yl acetate: [α]_D_^20^ = +30.1 (c = 0.073, CHCl_3_). IR (neat) ν_max_ 1797 (s), 1748 (s), 1237 (s) cm^−1^. ^1^H-NMR (700 MHz, CDCl_3_): δ 5.16 (1H, dd, *J =* 6.2, 6.2, H-3), 4.84 (1H, d, *J =* 6.5, H-2), 4.75 (1H, ddd, *J =* 5.7, 3.8, 3.8, H-4), 3.75 (1H, A part of an AB system further coupled, *J =* 12.0, 3.9, Ha-5), 3.65 (1H, B part of an AB system further coupled, *J =* 12.0, 3.6, Hb-5), 2.20 (3H, s, acetate). ^13^C-NMR: (175 MHz, CDCl_3_): δ 169.3, 168.7, 79.0, 70.9, 40.8, 30.0, 20.4. HRESIMS *m*/*z*: calcd for C_7_H_8_^79^Br_2_NaO_4_ 336.8687 [M + Na]^+^, found: 336.8692 [M + Na]^+^, 338.8671 [M + Na + 2]^+^, 340.8653 [M + Na + 4]^+^.

**15** (2R,3R,4S)-4-bromo-2-(bromomethyl)-5-oxotetrahydrofuran-3-yl acetate: [α]_D_^20^ = −22.4 (c = 0.23, CHCl_3_). IR (neat) ν_max_ 1796 (s), 1753 (s), 1212 (s) cm^−1^. ^1^H-NMR (500 MHz, CDCl_3_): δ 5.48 (1H, d, *J =* 3.5, H-3), 5.11 (1H, ddd, *J =* 9.0, 5.7, 3.5, H-4), 4.24 (1H, s, H-2), 3.64 (1H, A part of an AB system further coupled, *J =* 10.2, 5.6, Ha-5), 3.55 (1H, B part of an AB system further coupled, *J =* 10.2, 9.0, Hb-5), 2.17 (3H, s, acetate). ^13^C-NMR: (125 MHz, CDCl_3_): δ 169.5, 169.0, 78.6, 74.7, 38.3, 24.8, 20.4. HRESIMS *m*/*z*: calcd for C_7_H_8_^79^Br_2_NaO_4_ 336.8687 [M + Na]^+^, found: 336.8671 [M + Na]^+^, 338.8652 [M + Na + 2]^+^, 340.8631 [M + Na + 4]^+^.

**16** ((2R,3R,4S)-3-acetoxy-4-bromo-5-oxotetrahydrofuran-2-yl)methyl acetate: [α]_D_^20^ = −4.9 (c = 0.72, CHCl_3_). IR (neat) ν_max_ 1803 (s), 1741 (s), 1220 (s) cm^−1^. ^1^H-NMR (500 MHz, CDCl_3_): δ 5.39 (1H, dd, *J =* 3.2, 3.2, H-3), 4.64 (1H, m, H-4), 4.47 (1H, d, *J =* 3.5, H-2), 4.41 (2H, AB system further coupled; A part: *J =* 12.4, 3.9; B part: *J =* 12.4, 5.1, H_2_-5), 2.14 (3H, s, acetate), 2.12 (3H, s, acetate). ^13^C-NMR: (125 MHz, CDCl_3_): δ 170.2, 169.6, 169.5, 81.5, 76.9, 62.5, 38.7, 20.7, 20.5. HRESIMS *m*/*z*: calcd for C_9_H_11_^79^BrNaO_6_ 316.9637 [M + Na]^+^, found: 316.9643 [M + Na]^+^, 318.9621 [M + Na + 2]^+^.

**17** (S)-(4-bromo-5-oxo-2,5-dihydrofuran-2-yl)methyl acetate: [α]_D_^20^ = −65.9 (c = 0.42, CHCl_3_). IR (neat) ν_max_ 1776 (s), 1744 (s), 1227 (s) cm^−1^. ^1^H-NMR (500 MHz, CDCl_3_): δ 7.48 (1H, d, *J =* 1.9, H-3), 5.16 (1H, ddd, *J =* 4.7, 4.2, 1.9, H-4), 4.35 (2H, AB system further coupled: A part *J =* 12.2, 4.2; B part *J =* 12.2, 4.7, H_2_-5), 2.08 (3H, s, acetate). ^13^C-NMR: (125 MHz, CDCl_3_): δ 170.4, 167.5, 148.6, 115.0, 80.1, 62.1, 20.5. HRESIMS *m*/*z*: calcd for C_7_H_7_^79^BrNaO_4_ 256.9425 [M + Na]^+^, found: 256.9428 [M + Na]^+^, 258.9410 [M + Na + 2]^+^.

**18** (S)-3-bromo-5-(bromomethyl)furan-2(5H)-one: [α]_D_^20^ = −39.3 (c = 0.17, CHCl_3_). IR (neat) ν_max_ 1780 (s), 1609, 1151, 1033, 988 cm^−1^. ^1^H-NMR [20] (500 MHz, CDCl_3_): δ 7.59 (1H, d, *J =* 1.8, H-3), 5.20 (1H, ddd, *J =* 7.0, 4.5, 1.8, H-4), 3.67 (1H, A part of an AB system further coupled, *J =* 10.8, 4.5, Ha-5), 3.49 (1H, B part of an AB system further coupled, *J =* 10.8, 7.0, Hb-5). ^13^C-NMR: (125 MHz, CDCl_3_): δ 167.2, 150.1, 115.6, 80.0, 29.6. HRESIMS *m*/*z*: calcd for C_5_H_4_^79^Br_2_NaO_2_ 276.8476 [M + Na]^+^, found: 276.8471 [M + Na]^+^, 278.8452 [M + Na + 2]^+^, 280.8430 [M + Na + 4]^+^.

#### 3.3.1. Alcohol **19**

Pd/C (10% *w*/*w*, 4.5 mg) and Et_3_N (15 μL) were added to the alcohol **10** (22.5 mg, 0.0822 mmol) in EtOAc (3 mL), and the mixture was stirred under a hydrogen atmosphere. The same equipment used for the hydrogenation of **5** was used. After 1 h, the reaction mixture was worked-up as for **5**. Further filtration on a short pad of silica gel, eluting with EtOAc, gave crude **19**. HPLC purification on an analytical silica column (*n*-hexane-EtOAc, 3:7) afforded the pure alcohol **19** (12.8 mg, 76%) as an oil.

**19** (4S,5S)-5-(bromomethyl)-4-hydroxydihydrofuran-2(3H)-one: [α]_D_^20^ = −16.4 (c = 0.23, CHCl_3_). IR (neat) ν_max_ 3421 (br), 1776 (s), 1172 (s) cm^−1^. ^1^H-NMR (500 MHz, CDCl_3_): δ 4.61, 4.56 (1H each. m’s, H-3 and H-4), 3.61 (1H, A part of an AB system further coupled, *J =* 11.3, 3.9, Ha-5), 3.53 (1H, B part of an AB system further coupled, *J =* 11.3, 5.9, Hb-5), 2.99 (1H, dd, *J* = 18.3, 7.4, Ha-2), 2.58 (1H, dd, *J* = 18.3, 3.7, Hb-2). ^13^C-NMR: (150 MHz, CDCl_3_): δ 174.0, 84.8, 70.6, 37.8, 31.2. HRESIMS *m*/*z*: calcd for C_5_H_7_^79^BrNaO_3_ 216.9476 [M + Na]^+^, found: 216.9485 [M + Na]^+^, 218.9463 [M + Na + 2]^+^.

#### 3.3.2. Acetate **20**

A 65% HClO_4_ solution (two drops) was added to the alcohol **19** (7.0, 0.0359 mmol) in acetic anhydride (0.5 mL). After 1 h, the mixture was worked-up as usual. Purification by HPLC on an analytical silica column (*n*-hexane-EtOAc, 3:7) gave pure acetate **20** (6.4 mg, 75%), as a clear oil.

**20** (2S,3S)-2-(bromomethyl)-5-oxotetrahydrofuran-3-yl acetate: [α]_D_^20^ = −19.5 (c = 0.43, CHCl_3_). IR (neat) ν_max_ 1790 (s), 1741 (s), 1236 (s), 1162, 1041 cm^−1^. ^1^H-NMR (600 MHz, CDCl_3_): δ 5.25 (1H, d, *J* = 8.0, H-3), 4.73 (1H, bs, H-4), 3.74 (1H, A part of an AB system further coupled, *J =* 11.5, Ha-5), 3.65 (1H, B part of an AB system further coupled, *J =* 11.5, 3.8, Hb-5), 3.13 (1H, dd, *J =* 19.0, 8.0, Ha-2), 2.62 (1H, d, *J* = 19.0, Hb-2), 2.11 (3H, s, acetate). ^13^C-NMR: (150 MHz, CDCl_3_): δ 173.4, 170.4, 82.6, 72.6, 35.0, 32.2, 20.7. HRESIMS *m*/*z*: calcd for C_7_H_9_^79^BrNaO_4_ 258.9582 [M + Na]^+^, found: 258.9583 [M + Na]^+^, 260.9564 [M + Na + 2]^+^.

#### 3.3.3. Alcohol **21**

Pd/C (10% *w*/*w*, 3.0 mg) and Et_3_N (10 μL) were added to the alcohol **12** (11.9 mg, 0.0434 mmol) in EtOAc (6 mL) and the mixture was stirred under a hydrogen atmosphere. The same equipment employed for the hydrogenation of **5** was used. After 2.5 h, the reaction mixture was worked-up as usual. Further filtration through a short pad of silica gel eluting with EtOAc gave crude **21**. HPLC purification on an analytical silica column (*n*-hexane-EtOAc, 2:8) afforded pure **21** (3.4 mg, 25%) as an oil. 

**21** (4S,5R)-5-(bromomethyl)-4-hydroxydihydrofuran-2(3H)-one: IR (neat) ν_max_ 3419 (br), 1768 (s) cm^−1^. ^1^H-NMR (700 MHz, CDCl_3_): δ 4.73 (1H, bdd, *J* = 5.6, 3.8, H-3), 4.61 (1H, m, H-4), 3.69–3.62 (2H, m, H_2_-5), 2.84 (1H, dd, *J* = 17.9, 5.6, Ha-2) 2.65 (1H, d, *J* = 17.9, Hb-2), 2.4–2.0 (1H. broad, OH). ^13^C-NMR: (175 MHz, CDCl_3_): δ 174.3, 81.8, 67.7, 38.5, 26.3. HRESIMS *m*/*z*: calcd for C_5_H_7_^79^BrNaO_3_ 216.9476 [M + Na]^+^, found: 216.9472 [M + Na]^+^, 218.9451 [M + Na + 2]^+^.

#### 3.3.4. Acetate **22**

A 65% HClO_4_ solution (two drops) was added to alcohol **21** (2.6 mg, 0.0133 mmol) in acetic anhydride (0.5 mL). After 30 min, the mixture was worked-up as usual. Filtration through a short pad of silica gel gave crude **22** as an oil. Further purification by HPLC on an analytical silica column (*n*-hexane-EtOAc, 4:6) gave pure acetate **22** (1.1 mg, 35%), as an oil.

**22** (2R,3S)-2-(bromomethyl)-5-oxotetrahydrofuran-3-yl acetate: IR (neat) ν_max_ 1795 (s), 1735 (s), 1237(s) cm^−1^. ^1^H-NMR (700 MHz, CDCl_3_): δ 5.58 (1H, ddd, *J* = 5.8, 4.4, 1.1, H-3), 4.76 (1H, ddd, *J* = 8.3, 5.9, 4.3, H-4), 3.59 (2H, AB system further coupled, A part: *J =* 10.3, 5.9; B part: *J* = 10.3, 8.3), 2.93 (1H, dd, *J* = 18.3, 6.0, Ha-2), 2.67 (1H, dd, *J =* 18.3, 1.1, Hb-2), 2.13 (3H, s, acetate). ^13^C-NMR: (175 MHz, CDCl_3_): δ 173.0, 169.6, 80.3, 69.4, 36.8, 25.7, 20.7. HRESIMS *m*/*z*: calcd for C_7_H_9_^79^BrNaO_4_ 258.9582 [M + Na]^+^, found: 258.9569 [M + Na]^+^, 260.9550 [M + Na + 2]^+^. 

### 3.4. Synthesis of **18** with NaHSO_3_ (1.1 eq. or 1.5 eq.) in MeOH

NaHSO_3_ (3.9 mg, 0.0343 mmol, 1.1 eq.) was added to **8** (10.0 mg, 0.0317 mmol) in MeOH (1 mL) and the mixture was stirred at r.t. After 2 h, TLC analysis (*n*-hexane-EtOAc, 6:8) showed the complete conversion of **8** (*R*_F_ = 0.5) into **18** (*R*_F_ = 0.4). The reaction mixture was taken to dryness under an air stream and the residue was partitioned between water and CH_2_Cl_2_ (3 × 2 mL). The organic phase was dried, filtered and taken to dryness. Purification by HPLC on an analytical silica column (*n*-hexane-EtOAc, 75:25) gave pure **18** (2.5 mg, 30%) as an oil.

When the process was carried out under the same conditions (**8**: 9.8 mg, MeOH: 1 mL, NaHSO_3_ 1.5 eq.) for 22 h, compound **18** was obtained in a 25% yield.

### 3.5. Synthesis of **18** with NaHSO_3_ (10 eq.) in MeOH

NaHSO_3_ (83.8 mg, 0.805 mmol, 10 eq.) was added to **8** (25.3 mg, 0.0801 mmol) in MeOH (16 mL) and the mixture was stirred at r.t. After 3 h, the reaction mixture was worked up as above. Purification by HPLC (conditions as above) afforded **18** (9.9 mg, 45%).

### 3.6. Synthesis of **18** with NaHSO_3_ (1.2 eq.) in MeOH/H_2_O

NaHSO_3_ (10.6 mg, 0.102 mmol, 1.2 eq.) was added to **8** (26.9 mg, 0.0852 mmol) in MeOH/H_2_O (95:5, 4 mL) and the mixture was stirred at r.t. After 1.5 h, the reaction mixture was worked up as above. Purification by HPLC (conditions as above) afforded **18** (9.5 mg, 45%).

#### 3.6.1. Culture Conditions

Human cervix adenocarcinoma cell line (HeLa), malignant melanoma cell line (A375) and normal human fibroblasts (HDF) were grown in DMEM supplemented with 10% fetal bovine serum (FBS), 1% glutamine, 100 U/mL penicillin and 100 µg/mL streptomycin (Euroclone, Milano, Italy). Metastatic melanoma cell lines (WM266) were grown in RPMI medium supplemented with heat inactivated 10% fetal bovine serum (FBS), 2.5 mM glutamine, 100 U/ mL penicillin and 100 μg/mL streptomycin [25] (Euroclone). Cells were maintained in humidified air containing 5% CO_2_ at 37 °C. 

#### 3.6.2. Antiproliferative Activity

Cells were seeded at density of 2000 cells/well for WM266, 1200 cells/well for HeLa, 1000 cells/well for A375 and 2000 cells/well for HDF in 96-well microplates (Corning), with 100 µL of medium for well. After 24 h incubation, cells were treated with the substances at the indicated concentrations. The compounds were solubilized in DMSO at 50 mM concentration. Cell proliferation was revealed after 48 h treatment using the 3-(4,5-dimethylthiazol-2-yl)-2,5- diphenyltetrazolium bromide assay (MTT, Sigma Aldrich). Plates were then analyzed by using a microplate reader (Enspire, Perkin Elmer, Waltham, MA, USA) at 570 nm (MTT) [26]. The results are presented as the percentage of proliferating cells versus the control (DMSO treated cells) and are expressed as means ± SD of, at least, two independent experiments performed in triplicate. Statistical significance was determined by two-sided paired Student’s *t*-test. A *p* value less than 0.05 was considered significative. The IC_50_ values were calculated by GraphPad Prism software.

#### 3.6.3. Wound Healing Assay

In vitro cell migration has been evaluated using the wound healing scratch assay. A total of 6 × 10^5^ WM266 cells were plated and grown to the confluence. Successively, the cells were linearly scratched with a pipette tip to generate the wound. Once the detached cells were removed, the substance at indicated concentrations was added and the cells were incubated at 37 °C. After the scratch area was photographed at 0, 24 and 48 h using a phase optical microscope at 10× magnification (Zeiss), the distance between the edges of the incisions was measured [9]. Its mean value was determined as follows: wound closure (%) = (1 – wound width tx/wound width t0) × 100. Bars depict mean ± SE of three independent experiments.

## 4. Conclusions

In conclusion, taking mycalin A lactone as a model compound, we have synthesized several five-membered brominated lactones mimicking its lactone moiety. A SAR study revealed that the C-3 acetate compounds obtained from ribose, which possess an inverted configuration at C-3 with respect to mycalin A lactone at the corresponding stereocentre, are the most active and selective ones towards the tested tumor cells. Further studies performed on two of these substances, namely, compounds **18** and **20**, showed that they possess both high activity and selectivity towards the tested tumor cell lines and that they have a promising ability to inhibit the migration of WM266 cells. New conditions in which to obtain **18** in higher yields from an easily accessible material were found. All the other lactone acetates synthesized showed a similar high activity and selectivity, which justifies further studies on this interesting class of substance. The evidence collected also suggests possible synthetic modifications of both mycalin A and mycalin A lactone to access more active and selective related substances.

## Data Availability

The data presented in this study are available upon request from the corresponding author.

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
