# Peer review of "Synthesis of Brominated Lactones Related to Mycalin A: Selective Antiproliferative Activity on Metastatic Melanoma Cells and Inhibition of the Cell Migration"

_marinedrugs, 2023, doi:10.3390/md21060349_

Round 1
Reviewer 1 Report
This manuscript describes anti proliferative activity and inhibition of the cell migration of brominated pentose-derived lactones that were inspired from mycalin A lactone. Despite the simple structure of the molecules investigated, some of them (compounds 18 and 20) exhibit high anti proliferative activities and selectivities to tumor cells over than to human dermal fibroblast (HDF) cells. In addition, compound 20 showed significant inhibition of the cell migration. The reviewer thinks that these findings are intriguing and can be published in Marine Drugs. If the authors describe some information on the mechanism, for instance the target molecule, it would make the manuscript better.
Author Response
Actually we do not still know the mechanism, nor the target molecule (s). We will certainly perform other studies for substances like 18 and 20, as well as on the other more interesting lactones.
Reviewer 2 Report
In my opinion, the article presented by Piccialli et al is a very high-level paper that deserves to be published in Marine Drugs. However, there are some doubts as well as minor errors that must be corrected previously your approbation, which I detail below.
In the separation process of derivative 5, a mixture of this compound with derivative 7 is obtained. What is not clear to me is how both epimers could be separated, or were they used as a mixture. I think this should be explained in a better way in the writing, because by varying only one chiral center consider, in my opinion, that the purification process should not be easy, especially using conventional means such as HPLC. Or a chiral column was used in the purification?
In line 58 add separatory line.
In line 136 add parenthesis
In line 143 replace IC5o for IC50
In lines 170, 197, 223 and 251 add separatory line.
In line 437 add Aldrich dates (city, country, etc.)
Additionally, the authors must check the font type throughout the text.
Author Response
We thank this reviewer for his/her positive judgement of our manuscript. Epimeric compounds 5 and 7 have been separated by HPLC (well separated peaks have been obtained) on an analytical Phenomenex silica column using n-hexane-EtOAc (65:35). All the other errors have been corrected.
Reviewer 3 Report
The authors deal with synthesis and antiproliferative activity of new brominated lactones related to mycalin A, which is a polybrominated C15 acetogenin isolated from the encrusting sponge Mycale rotalis. I cannot assess the part of the work focused on the synthesis of substances, I'm not a synthetic chemist. For this reason, further is commented also only antiproliferative and antimigratory activity of tested compounds. The activity of the mycalin A, mycalin A acetate and mycalin A lactone and five bromolactone compounds mimicking the C1-C5 portion of mycalin A lactone was tested using human cancer cell lines including human melanoma (A375), human cervical adenocarcinoma (HeLa) and human metastatic melanoma (WM266) cancer cells. Tested compounds were very effective in these cancer cell lines. In contrast, one of these compounds, namely mycalin A lactone, was much less active on healthy human dermal fibroblast (HDF) cells. These results indicate the selective activity of mycalin A lactone for cancer cells. This selectivity is the main benefit of the mycalin A lactone compared to mycalin A and also mycalin A acetate, who also showed activity on healthy HDF and were therefore not selective for tumour cell lines. The presented structural analog of mycalin A, mycalin A lactone, retain the original antitumour activity but showed a reduced cytotoxicity towards healthy cells. It was also found that all the synthesized lactone acetates possess a cytotoxic activity comparable to, or higher than, that displayed by the parent lactone 3. Assays on the HDF cells, indicated that the C-3 epimeric lactone acetates derived from ribose displayed the highest selectivity towards tumour cells.
Also two of the above mentioned compounds were able to inhibit the migration of highly metastatic human melanoma WM266 cells during wound healing assay, which indicate the possible antimetastatic potential of these substances.
Thus, this result represents an advantage in comparison with the original mycalin A. These novel agents exhibit much better and selective cytotoxicity in cancer cell lines and also promising antimetastatic potential.
The manuscript is recommended for publication in the Marine Drugs.
Author Response
We thank this reviewer for his/her judgement on our paper.
Reviewer 4 Report
This manuscript desribes the synthesis of a large number of variously brominated and otherwise derivatized pentono-1,4-lactones inspired by the natural product mycalin A and the investigation of their anti-tumour efficiency to set up a structure-activity relationship. The chemistry follows well known lines, previous literature is properly cited. The work appears experimentally sound, the procedures are carefully described, the compounds are adequately characterized. A short discussion of the structural elucidation and differentiation +of the many stereoisomers would be useful. The judgement of the biological part is out of the expertise of this reviewer. The language of the manuscript is fine, only very minor improvements are needed (some hints are given below). On the other hand, the presentation of the results is not reader friendly, and it would be very useful to include a graphical representation of the SAR with structural formulae. I am quite unhappy by the structure of the manuscript, namely the mix-up of the synthetic and biological parts; it is strongly suggested to separate these (even if the work was carried out chronologically as presented), to describe the chemistry first for all compounds, then the antiproliferative studies; in addition all descriptions are very lengthy, the results should be presented in a more concise way. The work deserves publication in Marine Drugs after major revision that takes into account also the points listed below.
Detailed corrections and suggestions
the configurational descriptors D should be in lower case small capitals according to the IUPAC carbohydrate nomeclature
throughout the text - „based on reacted starting material, borsm” – this concept is known as the conversion of the starting material, and the yield can be based on the total amount of the reacting material or this can be corrected by the conversion; I think this would better correspond to the conventions
l41 delete synthetic
Fig 1 pls, indicate also in the drawing (not only in the legend) for the right box that these are the target compounds of the study; I think, α/β are not proper denotions here (in carbohydrate derivatives they refer to the anomeric configuration), suggest to use wavy lines instead
l60-62 pls, add some kind of evidence for the assumption that the lactone portion of 3 is responsible for the activity, e. g. has this lactone part alone been tested? (otherwise this is not a rational basis of the compound design) – in lines 80-81 it is stated that the purpose of this study was to clarify „whether the lactone moiety in 3 might be responsible for its activity”; please, make these assumptions/background informations clear and better readable/understandable, e. g. by inserting more structural formulae and biological data into Fig. 1
l62 actually, not the commercial compounds were used in the study as they are described in the experimental; at least in the first mention, pls, use the name of the starting lactones that also reflects the ring size (e. g. D-xylono-1,4-lactone)
l70-74 this paragraph would fit rather in the conclusion part, should be deleted here
Scheme 1 HPLC is not a reagent or reaction condition, should be deleted from the scheme; pls, unify the appearance of the conditions by mentioning reaction time and temperature everywhere
l92 compd 6 seems also to be known from ref 12
l107 I think the yield is not meaningful here, as it is not known what would happen after longer reaction time, would the starting compound be fully converted or an equilibrium would be reached, the given percentage reflects rather the separation than the reaction itself
Scheme 2 and section 3.2.5 why was the reaction interrupted after 1 h, why the full conversion was not waited for?
Fig. 3 please, indicate in the panels which numbers refer to the acetates and which to the alcohols; would it be possible to turn the compound numbers to a standing position?
Fig. 3 and l136 I see no reason why the alcohols’ data on HDF are not shown here (it would be more reader friendly to show the comparisons in one and the same place);
Fig. 4 it would be extremely reader friendly to insert the structural formulae of the investigated compounds into each panel of the figure
l175-183 α/β are also misused in this paragraph, here the usage of corresponding descriptors of the relative configuration would be appropriate
Fig. 5 this is a useful summarizing presentation of the compounds, nevertheless, I suggest to include all of the synthesized test compounds that would make finding a structure easier for the reader
l302-303 it would be reader friendly to make a brief statement about what can be expected in the SI about these substances
l420 diaxialiy
l450 pls, indicate the solvent
l487 yield for 7, pls, check Scheme 1 (1%) vs 3% here
l517 was added
calling the compounds obtained from D-ribonolactone as ribose-derived ones is correct historiographically, is, however, misleading when used in the analysis of the structure-activity relationships, as they are mostly no more D-ribo configured (this is also valid for the xylose-derived ones); also in the experimental (subtitles in l458, l563)
Experimental
for the many identical preparations the use of general procedures would be welcome
l445 pls, check the measuring frequencies, several times 400 and 500 MHz are indicated in the descriptions, compare also the SI
compounds’ names according to the IUPAC carbohydrate nomenclature should be added
in the MS data the molecular ion clusters should be indicated to prove the presence of one or more Br atoms
l568 one third of the reaction mixture… what was done with the rest?
l655 75%
subsections 3.4-3.6 pls, keep the most efficient only, variations appear in Table 4
l753 ff pls, indicate only that in the SI copies of the NMR spectra are collected, it is not needed to list them in details here
References
doi-s are missing
ref 13 may be moved to the Acknowledgement section
Supplementary Information
please, insert the structural formulae into the NMR sepctra
Essentially fine, minor corrections can be done in the proof.
Author Response
Corrections according to reviewer 4. Point-by-point response follow.
First of all we thank this reviewer for his/her effort in reading and correcting our paper.
All the corrections/modifications have been highlighted in yellow.
The Introduction has been rearranged in a more reader friendly manner by including in figure 1, as suggested by this reviewer, a graphic of the antiproliferative activity of mycalin A, its acetate and the lactone derivative.
Figure 2 has been added, where the formulae of the compounds investigated in this paper have been shown, by also adding in the figure the following specification: “Compounds investigated in this study". Wavy lines have been used in place of a/b. Some repetitions have been deleted as well.
According to IUPAC nomenclature, the descriptors D have been changed to lower case small capitals throughout the paper.
A brief description of some salient nmr features of each lactone has been added.
As for the yield of compound 8 which we report as “borsm”, we have calculated it based on the reacted 5. I do not understand where the error is. Could this reviewer explain his/her observation in a clearer manner?
Line 41. “synthetic” deleted.
Lines 60-62. The assumption that the lactone portion in 3 could be responsible for the activity/selectivity of this substance rests on the evidence highlighted in lines 39-44 where is reported that “various analogues of mycalin A were synthesized” (ref.7). and also, that the biological assays carried out on these compounds showed that “the THF-containing or THF-mimicking portion of mycalin A is essential to biological activity”.
Line 62. “Commercially available” deleted. D-xylono-1,4-lactone and D-ribono-1,4-lactone in place of D-xylonolactone and D-ribonolactone.
Lines 70-74. This paragraph has been deleted. This information is already given in the conclusions.
Lines 75-78. This sentence has been rewritten: the same concept had already been conveyed on lines 59-60. In addition, as suggested by this reviewer, a summary of all the D-xylonolactone-derived compounds has been included in Figure 2. Meanwhile, Mycalin A lactone has been deleted in Figure 2.
Scheme 1. “HPLC” on the arrow has been deleted. A brief sentence has been added on line 96: “compounds 5 and 7 were separated by HPLC”. Also, the yield of 7 has been corrected to 3%.
Reaction times and temperatures have been specified for every reaction.
Line 92. Indeed, compound 7 is known, but, as we specified on lines 96-99, the chemical shift value of some signals was different from that reported and we preferred to confirm the structure of this compound by X-ray diffraction analysis, also considering its great tendency to crystallize.
Line 107 and Scheme 2. Indeed, we do not know if this is an equilibrium mixture or if it reflects the amounts obtained after one hour reaction. We simply stopped the process after 1 h, when the process appeared not to further proceed. We have deleted the yield both on line 107 and in Scheme 2.
Fig. 3. In the panels it has been specified which numbers refer to the acetates and which ones refer to the alcohols.
Fig 3 and line 136. Having observed that the alcohols 5, 6 and 7 were essentially inactive against tumor cells, we did not measure their activity on HDF cells. We have also deleted: “For the alcohols see S1”. This was an error.
Line 175-183. Descriptors R/S have been used.
Line 420. Corrected with “diaxially”.
Line 450. CDCl3 added.
Line 487. 3% is right here, in agreement with Scheme 1, corrected as above.
Line 517. “was added”.
Lines 458 and 563. “Xylose-derived” and “ribose-derived” have been deleted.
“Ribose-derived” and Xylose-derived have been either deleted or replaced by D-ribonolactone-derived and D-xylonolactone-derived, throughout the paper.
Experimental
Indeed compounds 4, 8 and 9 have been prepared using the same reaction. However, the procedures are slightly different about quantities, reaction time, TLC Rf values, appearance of the product, HPLC purification conditions, all data that we believe are worth reporting. We have reduced to a minimum each recipe. For compounds 19, 20, 21 and 22 the recipe has been reduced further.
Line 568. On line 577 it is specified that “MeOH was added in portions to the remaining original reaction mixture”.
Line 655. 75% corrected.
Line 753. As suggested, we have simplified the Supporting Materials section by only specifying “copies of NMR spectra are available”.
References. Doi are not required by the journal. They add “Cross ref” in the editing phase.
In the Supplementary Section the formulae have been added into the NMR spectra.
Finally, we understand the preference of this reviewer to assemble the manuscript by first describing the chemistry and then the biological activity. However, we believe that this is matter of style. On the other hand, it would be very difficult for us to rewrite the entire paper. We hope this reviewer could understand our point of view.
Round 2
Reviewer 4 Report
The authors have properly addressed most of the remarks.
What is still missing:
11. in the figures - would it be possible to turn the compound numbers to a standing position (in some places this has been managed)?
22. pls, check the NMR measuring frequencies indicated in the general methods, several times 400 and 500 MHz are indicated in the descriptions, compare also the SI
33. compounds’ names according to the IUPAC carbohydrate nomenclature should be added in the experimental
44. I still hold my view on the structure of the manuscript (no mixing of chemical and biological results), nevertheless, would leave this point to the decision making editor
minor corrections to be carried out
Author Response
Compounds' numbers in figure 4 have been turned to a standing position
The NMR measuring frequencies have been cecked as well as the correspondence with those reported in the supplementary material.
THe compounds’ names, according to the IUPAC nomenclature, have been added in the experimental section.